# A Novel ML-Powered Nanomembrane Sensor for Smart Monitoring of Pollutants in Industrial Wastewater

**DOI:** 10.3390/s25175390

**Published:** 2025-09-01

**Authors:** Gabriele Cavaliere, Luca Tari, Francesco Siconolfi, Hamza Rehman, Polina Kuzhir, Antonio Maffucci, Luigi Ferrigno

**Affiliations:** 1Department of Information and Electrical Engineering and Applied Mathematics, University of Salerno, 84084 Fisciano, Italy; 2Department of Electrical and Information Engineering, University of Cassino and Southern Lazio, 03043 Cassino, Italy; luca.tari@unicas.it (L.T.); francesco.siconolfi@unicas.it (F.S.); maffucci@unicas.it (A.M.); ferrigno@unicas.it (L.F.); 3EUT+ Institute of Nanomaterials and Nanotechnologies-EUTINN, European University of Technology, European Union, 03043 Cassino, Italy; 4Department of Physics and Mathematics, Center for Photonics Sciences, University of Eastern Finland, 80100 Joensuu, Finland; hamzar@student.uef.fi (H.R.); polina.kuzhir@uef.fi (P.K.)

**Keywords:** environmental monitoring, pollutant detection, smart monitoring, data analysis, machine learning, nanosensor, industrial wastewater

## Abstract

This study presents a comprehensive analysis aimed at validating the use of an innovative nanosensor based on graphitic nanomembranes for the smart monitoring of industrial wastewater. The validation of the potential of the nanosensor was carried out through the development of advanced analytical methodologies, a direct experimental comparison with commercially available electrode sensors commonly used for the detection of chemical species, and the evaluation of performance under conditions very similar to real-world field applications. The investigation involved a series of controlled experiments using an organic pollutant—benzoquinone—at varying concentrations. Initially, data analysis was performed using classical linear regression models, representing a conventional approach in chemical analysis. Subsequently, a more advanced methodology was implemented, incorporating machine-learning techniques to train a classifier capable of detecting the presence of pollutants in water samples. The study builds upon an experimental protocol previously developed by the authors for the nanomembranes, based on electrochemical impedance spectroscopy. The results clearly demonstrate that integrating the nanosensor with machine-learning algorithms yields significant performance. The intrinsic properties of the nanosensor make it well-suited for potential integration into field-deployable platforms, offering a real-time, cost-effective, and high-performance solution for the detection and quantification of contaminants in wastewater. These features position the nanomembrane-based sensor as a promising alternative to overcome current technological limitations in this domain.

## 1. Introduction

Industrial wastewaters are water streams contaminated during manufacturing or processing activities. Commonly identified contaminants in such effluents include heavy metals (e.g., lead, mercury), organic compounds, suspended particulate matter such as oils and greases, and various synthetic chemical agents [1,2,3,4].

In the absence of adequate treatment, industrial wastewater discharge can pose serious risks to human health and ecosystems [5].

Consequently, the proper treatment and management of industrial effluents before environmental release is paramount to safeguarding public health and maintaining ecological integrity. To this end, industrial facilities are typically equipped with WWTPs [6,7] which, through purification processes, enable the partial or complete removal of contaminants in the wastewater before discharge or reuse.

Regulatory frameworks mandate that industrial operators conduct systematic wastewater monitoring, both before and after treatment, to verify compliance with permissible pollutant levels. These legal requirements, which vary depending on jurisdiction and the nature of the industrial process, are enforced through an intricate set of international, national, and regional environmental protection policies [8].

Monitoring waterborne pollutants is strategically critical for industries for two primary reasons:To secure and maintain discharge permits by evidencing adherence to environmental standards;To proactively assess effluent quality, reducing the risk of non-compliance penalties during unannounced regulatory inspections.

A wide variety of technological solutions have been explored in the field of pollutant monitoring and industrial applications. For instance, novel image-based monitoring approaches have been developed to detect and analyze air pollution originating from industrial emissions [9]. In parallel, nanomaterial- and membrane-based strategies have been investigated for industrial separation processes and water treatment. For example, MXene-based composite membranes incorporating Na-Bentonite have demonstrated excellent separation performance in purification processes [10]. In contrast, covalent organic framework membranes fabricated through scalable and environmentally friendly techniques such as scraping-assisted interfacial polymerization have emerged as promising candidates for large-scale water purification [11]. Furthermore, studies on adsorption and mass transfer in porous carbon-based nanomaterials have opened new pathways for CO_2_ capture and environmental remediation [12].

In the industrial domain, the development of advanced sensing technologies based on AI has also played a crucial role. For instance, self-supervised deep-learning frameworks, such as the multi-head attention self-supervised model, have been proposed for intelligent feature extraction from sensor data in manufacturing maintenance [13]. Vision-based inspection techniques have similarly advanced, with multi-label defect classification methods designed to enhance the reliability of sewer inspection systems [14]. In the context of water quality, other monitoring systems have been reported, including aptamer-based colorimetric assays integrated with centrifugal microfluidic chips for the simultaneous detection of Pb^2+^, Hg^2+^, and As^3+^ in water [15].

Despite these significant advances, intrinsic limitations remain that hinder large-scale or long-term deployment. Specifically, challenges persist in achieving continuous monitoring, direct in situ pollutant detection, and cost-effectiveness. This underscores the urgent demand for innovative and reliable technological solutions simultaneously satisfying three essential requirements: real-time water quality assessment, direct in situ pollutant detection, and low-cost instrumentation. According to [16] Table 1 summarizes the performance of each sensor family in terms of real-time capability, portability and limitations.

Beyond this review, several other studies have introduced alternative sensor technologies for water quality monitoring. While these approaches have provided significant scientific advances and novel applications, they also exhibit intrinsic limitations restricting their large-scale or long-term deployment. A comparative overview, proposed by the authors, of these technologies and their main constraints is reported in Table 2.

The present study is framed on these regulatory and technological imperatives, building upon the work presented in [49] by the authors. Specifically, it thoroughly examines the application of an innovative single-use nanosensor, previously described in [49], to detect waterborne pollutants.

The nanosensor, a graphitic nanomembrane, operates by evaluating response variations upon interaction with the target chemical species, applying EIS. This sensing approach, leveraged alongside the intrinsic properties of the nanomembrane, facilitates the development of innovative smart monitoring systems. In particular, these nanosensors demonstrate significant potential for implementing real-time analysis with low-cost hardware platforms and directly in situ, potentially addressing the described limitations in existing technologies.

The main contribution of this work lies in the introduction of a disposable graphitic nanomembrane sensor combined with ML-powered analytical methods which, unlike existing monitoring technologies, has the potential to simultaneously fulfil the three critical requirements for wastewater pollutant monitoring—real-time responsiveness, in situ applicability, and cost-effective implementation—while ensuring reliable performance. At the present stage, the authors focus on exploring the potential of nanomembranes for the future development of real-time and in situ smart monitoring systems, evaluating their performance as a foundational step. Accordingly, the objective of this study is to validate the feasibility of the nanomembrane sensor for smart water monitoring in industrial contexts, through three distinct phases:
Establish a new analysis methodology by exploiting the nanomembrane electrochemical properties;Evaluate performance in terms of sensitivity and selectivity via experimental comparison with commercial electrode sensors and conventional chemical analysis techniques;Evaluate performance in terms of sensitivity and selectivity under varying operational conditions for real-world field applications.

Regarding the first step, a comprehensive analysis of the physicochemical properties of the nanomembranes was conducted. This investigation enabled the identification of novel figures of merit to support the development of an analytical framework for water quality monitoring.

Concerning the second step, a series of experimental tests was carried out using water samples containing varying concentrations of an industrial organic contaminant, benzoquinone. These tests were performed on both nanomembrane-based sensors and commercial sensor platforms. Performance was evaluated before by assessing the contaminant concentration estimation capability (selectivity), and in a second stage, by evaluating detection capability—namely, the ability to discriminate between contaminated and uncontaminated samples (sensitivity). Given that the experimental comparison involves both hardware (i.e., different instruments and sensors) and software (i.e., distinct analysis methodologies), an initial assessment was conducted using conventional analytical techniques. In particular, linear regression models were employed for both sensors’ outputs. The results indicated that the application of linear models did not enhance the properties of nanomembranes, nor did it produce satisfactory performance. These results highlight the need to adopt alternative and more advanced analytical approaches better suited to the nanosensor outputs. Consequently, the analysis was extended by integrating ML algorithms to determine whether advanced data-driven methods could enhance performance with respect to linear models, demonstrating accuracies above 94% for concentration estimation and 100% for detection capability—values comparable to those achieved with established traditional electrochemical techniques (99.7% and 99.9%, respectively). The ML techniques applied included Support Vector Machines, k-Nearest Neighbors, Logistic Regression, and Naive Bayes classifiers. The comparative use of different models was necessary to develop an analysis protocol for nanomembrane-based sensing, as not all models lend themselves well to nanosensor output. It is important to note that this stage aims not to assert the superiority of the nanomembrane sensor over existing commercial solutions. Instead, it is to demonstrate that, despite the prototypal stage, the sensor achieves performance levels comparable to established technologies. This positions it as a credible and innovative alternative for real-time, in situ environmental monitoring applications.

Finally, in alignment with the third step, an in-depth analysis was performed to assess the optimal test time and robustness under noisy conditions, introducing Gaussian noise at varying intensities to the raw nanosensor data and analyzing the resulting impact on performance. These aspects were evaluated to gauge the feasibility of deploying the nanosensor in real-time and in situ operational contexts.

Based on these considerations, the structure of the paper is organized as follows: Section 2 provides a detailed description of both sensor typologies and the analytical methodologies employed; Section 3 outlines the experimental protocols and preliminary findings; Section 4 and Section 5 present a comparative performance evaluation and a deeper technical analysis of the nanomembrane-based sensor; and finally, Section 6 concludes the study with key insights and future research directions.

## 2. Sensors and Methods

This section comprehensively describes the innovative single-use nanosensor developed within the research framework presented in [49,50,51] and the conventional sensor based on commercial electrode-based sensors. We detail the electrochemical techniques used to characterize contaminated water samples for each sensor type and the related analytical approaches. All described methods are suitable for evaluating the electrochemical properties of dissolved chemical species and are directly applicable to the analysis of industrial wastewater.

### 2.1. Electrode Sensors and Conventional Methods

The commercial sensor selected for this study, detailed in [52], is the ItalSens Carbon SPE. It consists of a three-electrode configuration incorporating two graphite electrodes and a silver reference electrode, with a working electrode area of 0.07 cm^2^. Figure 1a shows an image of the employed SPEs. These sensor electrodes are well-suited for chemical analysis using strongly consolidated conventional electrochemical methods.

The techniques employed for performance evaluation included cyclic voltammetry, chronoamperometry, and electrochemical impedance spectroscopy. All measurements were conducted using a commercially available instrument specifically designed for chemical sensing applications, and the applied methodologies are fully compliant with pollutant analysis.

Conventional figures of merit were extracted for each technique to assess the sensor’s capability to detect and quantify the target analyte.

In CV [53], a triangular voltage waveform is applied, and the resulting current is acquired. The anodic and cathodic peak currents exhibit a linear correlation with analyte concentration, as governed by the Randles–Sevcik equation, provided the redox process is reversible or quasi-reversible. Furthermore, since the peak current, at parity of chemical species, is empirically related to the area of the entire curve, it is possible to observe the latter directly. Since these considerations hold true only if the redox reactions involving the pollutant–electrode sensor system are reversible or quasi-reversible, a preliminary characterization was conducted to assess the reversibility of the electrochemical processes. The results confirmed that the system exhibited no signs of irreversibility.

In CA [54], a constant potential step is applied, and the resulting transient current is monitored over time. The current decay profile conforms to the inverse square root behavior characteristic of the Cottrell equation, which describes diffusion-controlled reactions. The steady-state current plateau can be plotted against analyte concentration to construct a calibration curve, as Faradaic processes ensure a linear current-concentration relationship.

For EIS [55], an AC signal is applied across a defined frequency range, and the impedance response is recorded for different pollutant concentrations. Standard data representations include both Bode and Nyquist plots; in this study, particular emphasis is placed on the latter to evaluate the charge transfer resistance, inversely proportional to analyte concentration.

### 2.2. Nanosensor and Non-Conventional Method

The novel sensor investigated in this study consists of a nanostructured membrane derived from Pyrolysed Photoresist Films doped with nickel. Such an approach allows for converting several hundred nanometers-thick photoresist layers into ultrathin graphitic films with thicknesses in the order of tens of nanometers, embedded with Ni nanoparticles. This process enables scalable and reproducible fabrication of graphitic films, with an investigated relationship between synthesis conditions and resulting material properties [50]. As described in [49,50,51], the membrane is fabricated via spin-coating 300 µL of nLoF resist (AZR 1:4) at 3000 rpm for 60 s onto a SiO_2_ wafer pre-coated with a 10 nm nickel catalyst layer. The samples are then baked at 110 °C for 1 min and subsequently heated from 100 °C to 700 °C at 20 °C/min in an inert atmosphere, followed by a slower ramp from 700 °C to 800 °C at 10 °C/min. Finally, they are annealed at 800 °C for 30 min and cooled overnight in a static hydrogen atmosphere.

The photoresist film pyrolysed in the presence of a Ni layer was transformed into a graphitic film, while the molten Ni film broke up into an array of micro- and submicron-sized nickel particles. The resulting carbon membrane had a measured thickness of 60 nm. Carbon films were investigated by Zeiss LEO 1550 SEM (Carl Zeiss AG, Oberkochen, Germany) to check the integrity of the films. As shown in Figure 2a, the PPF exhibits a uniform surface. The embedded submicron nickel particles are revealed in the high-resolution SEM images in Figure 2b. Notably, the presence of these nickel particles may increase the surface roughness of the film compared to PPF [50].

The distinguishing feature of this sensor is its two-dimensional nanomembrane architecture, which provides a high surface-to-volume ratio, greatly enhancing its molecular adsorption capacity.

In contrast to traditional electrode sensors, which operate through redox reactions, the nanosensor’s transduction mechanism is based on permanent structural deformations within the membrane’s lattice upon interaction with external molecules. This deformation arises from minimizing the system’s total energy as atoms rearrange in response to external species, which results in a measurable change in electrical conductivity. To quantitatively describe this mechanism, a simplified model for the electron transport in nanoscale conductors is provided by the following generalized Ohm’s law that relates the current density ${\hat{J}}_{z}$ to the electric field ${\hat{E}}_{z}$ in the wavenumber, *k*, and frequency, ω, domains [56]:(1)$${\hat{J}}_{z}\left(k,\omega \right)=\frac{\sigma_{0}}{1+j\omega \tau }\;\cdot \;\frac{1}{1-\psi \left(\omega \right)k^{2}}\;\cdot \;{\hat{E}}_{z}\left(k,\omega \right)$$
with:(2)$$\psi \left(\omega \right)=\frac{\xi \left(\omega \right)v_{F}^{2}}{\nu^{2}{\left(1+\frac{j\omega }{\nu }\right)}^{2}}$$
and:(3)$$\sigma_{0}=\frac{2v_{F}M}{\nu R_{0}X}$$

Here, $\sigma_{0}$ is the DC conductivity given by Equation (Equation 3), where $R_{0}$ = 12.9 kΩ is the quantum resistance, *M* is the number of conducting channels, $v_{F}$ is the Fermi velocity, ν is the collision frequency, and τ = 1/ν is the relaxation time. In addition, the quantities *X* and ξ(ω) depend on the material. For instance, for a metallic nanowire of diameter *D*, it is *X* = π(*D*/2)^2^, whereas ξ(ω)≈(1+1.8jω/ν)/3(1+jω/ν) [56].

Note that Equation (Equation 1) implicitly defines a complex impedance in the wavenumber domain, whose parameters can be strongly affected in the case of absorbance of external molecules, both in the case of physical and chemical interaction. A physical interaction can occur when the external substances are simply trapped in the nanomaterial lattice: in this case, the deformation of the lattice properties (for instance, the local distance between atoms) gives rise to changes in the collision frequency ν, and in the number of conducting channels *M* [57]. In case of chemical interaction between the nanomaterial and the external molecules, new bonds are created, so that also the describing functions ξ(ω) will be affected [58].

The 2D nature of the nanomembranes amplifies the sensitivity to this phenomenon, as the whole material is exposed to the absorption of external substances.

In the general case, a time-domain formulation of Equations (Equation 1) and (Equation 2) may only be evaluated by numerical inversion from the space and frequency domains. However, in the low spatial frequency limit, a time-domain formulation can be cast in the frame of the transmission line theory, linking the time and space variations of voltages and currents through the following model [59,60]:(4)$$-\frac{\partial V(t,z)}{\partial z}=R_{TL}^{\prime }I\left(t,z\right)+L_{TL}^{\prime }\frac{\partial I(t,z)}{\partial t}$$(5)$$-\frac{\partial I(t,z)}{\partial z}=C_{TL}^{\prime }\frac{\partial V(t,z)}{\partial t}$$
where the per-unit-length parameters $R_{TL}^{\prime }$, $C_{TL}^{\prime }$ and $L_{TL}^{\prime }$ are expressed in terms of the quantum parameters defined in Equation (Equation 2) and in terms of the classical electrical capacitance and magnetic inductance. Solving Equations (Equation 4) and (Equation 5) allows following the time evolution of the relation between voltages and currents.

In [49], a preliminary metrological characterization of the nanosensor was performed, encompassing stability, repeatability, and sensitivity assessments.

The non-conventional measurement strategy proposed in this work employs time-domain EIS, which enables continuous sensor monitoring both in the presence and absence of target pollutants. Unlike conventional EIS—where measurements commence only after sample deposition—the nanomembrane sensor allows for uninterrupted operation even without a conductive medium, due to its intrinsic, redox-independent transduction mechanism.

The interaction mechanism between the nanosensor and external substances, based on permanent deformations in the lattice of the nanomaterial, enables the development of novel analytical methodologies. One such approach involves observing the sensor’s differential response relative to a reference condition—specifically, the absence of external substances. The reference condition will hereafter be referred to as the baseline. This capability represents a key distinction from electrochemical impedance spectroscopy performed on traditional electrode sensors, where differential responses cannot be observed. In the absence of an external solution, which is essential for electron conduction between electrodes, the system behaves as an open circuit due to the lack of a conductive medium. As a result, any acquired signal would be dominated by noise and lack physical significance. The ability to register meaningful differential responses concerning the baseline conditions is therefore a distinctive and novel feature of this nanosensor, setting it apart from conventional electrode-based sensors.

For data interpretation, while conventional EIS typically considers both the magnitude and phase of the impedance—analyzing the real and imaginary components—the evaluation of the nanosensor focuses primarily on the impedance magnitude. This is because preliminary tests have shown that, in this specific context, the interaction with external molecules does not induce significant phase variations. Moreover, the impedance phase contributes minimally (approximately 1°), making the changes in impedance magnitude the most meaningful and representative aspect of the nanosensor’s differential response.

Moreover, in this work, a differential analysis strategy is proposed. Instead of absolute impedance values, sensor response is evaluated by impedance variation relative to the pre-exposure reference condition, previously defined as baseline. The following equation gives the normalization process:(6)$$M_{norm}\left(f_{i},t_{i}\right)\left[\%\right]=\frac{M_{abs}\left(f_{i},t_{i}\right)-B\left(f_{i}\right)}{B(f_{i})}\;\cdot \;100$$
where:$M_{norm}\left(f_{i},t_{i}\right)$ is the normalized nanomembrane impedance module at a given frequency $f_{i}$ and at time $t_{i}$, expressed in percentage;$M_{abs}\left(f_{i},t_{i}\right)$ is the absolute nanomembrane impedance module at a given frequency $f_{i}$ and at time $t_{i}$;$B(f_{i})$ is the baseline nanomembrane impedance module at frequency $f_{i}$, calculated as the average value in a specific time window before the drop.

Given the variability in absolute impedance across custom-fabricated membranes, these normalized responses ensure consistency across different membrane samples and analyte concentrations. The normalization ultimately allows for the observation of a trend between the nanomembrane responses and the tested concentrations that, looking only at the absolute impedance modulus, would not have been possible to obtain.

For each excitation frequency, impedance variation is computed along with a secondary figure of merit—defined in Equation (Equation 7)—which quantifies the spread of normalized impedance values after the external molecules drop.(7)$$s_{F}\left(t_{i}\right)\left[\%\right]=\frac{\max\langle M_{abs}\left(t_{i}\right)\rangle -\min\langle M_{abs}\left(t_{i}\right)\rangle }{\mathrm{mean}\langle M_{abs}\left(t_{i}\right)\rangle }\;\cdot \;100$$
where:$s_{F}\left(t_{i}\right)$ represents the nanomembrane Spread Factor at time $t_{i}$ after the drop;$M_{abs}\left(t_{i}\right)$ is the absolute nanomembrane impedance module at time $t_{i}$ after drop, where maximum, minimum and average values are evaluated over all frequencies.

These figures of merit fundamentally differ from those used in classical EIS analysis and jointly define an innovative analytical feature space for quantifying pollutants.

## 3. Experimental Setup and Preliminary Results

This section provides a detailed account of the experimental procedures and instrumentation employed to validate the use of the nanomaterial as a water quality sensor. Preliminary test results are also presented, emphasizing key aspects relevant to the analyses discussed in the subsequent sections. In particular, this section highlights the necessity of normalizing the responses of the nanomembranes as part of the data analysis.

### 3.1. Test Setup

The pollutant selected for the experimental investigation is benzoquinone, a hazardous organic compound commonly encountered as a by-product in various industrial processes. In this study, benzoquinone was dissolved in a 0.1 M acetate buffer solution at pH 4, supplemented with 0.1 M KCl. This chemical environment was selected to match the current development stage of the graphite-based nanomembranes, which have not yet been functionalized and are therefore not yet suitable for real wastewater samples. Functionalization—essential for real-world applications—enables selective analyte binding and minimizes interference from complex chemical species matrices. The development of selective coatings is currently the subject of ongoing research by the authors.

To the work aim, four concentration levels were investigated: 0 mM (i.e., pure water), 0.1 mM, 1 mM, and 10 mM. Although the actual occurrence levels of benzoquinone in industrial wastewater are not well documented in the literature—given that it is classified as an “emerging” organic pollutant arising from aromatic oxidation processes (e.g., phenol degradation [61,62])—the selected concentration range remains consistent with values typically explored in experimental studies [63,64,65]. The upper limit of 10 mM, while likely higher than concentrations expected in real-world scenarios, was intentionally chosen as a limit case, serving as a figure of merit for benchmarking the nanosensor’s performance (i.e., operating range). This approach ensures characterization across a broader dynamic range than what may be encountered in practice, thereby strengthening the evaluation of the nanosensor’s detection capabilities under both realistic and extreme conditions.

For each concentration, four independent sensors were tested, yielding sixteen sensors per technology, both for nanomembranes and commercial electrode-based sensors, enabling comparative analysis. For the conventional electrochemical techniques applied to commercial SPEs, the following experimental protocols were adopted:CV: A potential sweep from −1 V to +1 V was applied twice using a triangular waveform. The potential increased by 0.01 V at each measurement, with a scan rate of 0.05 V/s. Voltage, anodic, and cathodic currents were extracted at the second sweep for analysis, for a total of 80 s each test.CA: A potential step of −0.05 V, consistent with the CV results, was applied. The resulting current was recorded over 3 min, yielding 1800 data points per SPE.EIS: A sinusoidal forcing potential of 0.01 V was applied, in “time scan” mode. Measurements were performed over a total duration of 15 min per SPE, with 20 complete frequency sweeps logarithmically spaced between 20 Hz and 1 MHz, for a total of 35 frequencies.

Each of these methods was performed after dropping a drop of pollutant with a volume of 80 μL on the WE of the SPE.

A continuous time-domain EIS monitoring protocol was adopted for the nanostructured graphite-based membranes, lasting 20 min per test. The first 5 min were used for baseline acquisition, as detailed in Section 2, with the sensor exposed to ambient air. Subsequently, 500 μL of benzoquinone solution at the target concentration was dispensed directly onto the membrane surface using a precision micropipette.

### 3.2. Experimental Setup

Electrochemical measurements on commercial electrode-based sensors were conducted using a PalmSens4^TM^ instrument [66] (PalmSens, Houten, The Netherlands), connected to a PC via USB. The system supports all applied techniques—CV, CA, and EIS—with acquisition managed via the PSTrace (PalmSens, Houten, The Netherlands, 5.9 version) software. For the commercial electrodes used with the PalmSens, a preliminary optimization phase preceded the start of each test for each technique, following the standard protocol reported in the manual of the instrument for this type of chemical measurement. The complete configuration for these tests is shown in Figure 3a.

GW Instek LCR meter (Good Will Instrument Co., Taipei, Taiwan) was employed for time-domain EIS measurements on nanomembranes, interfaced with a computer via the GPIB-488 protocol, as illustrated in Figure 3b. Calibration (open and short) was performed according to the manufacturer’s specifications to minimize parasitic and environmental noise. The system operated in SLOW acquisition mode using a 2 V AC excitation signal, chosen to balance sensitivity with the need to avoid damaging the nanomembranes. This is thanks to previous characterization [49]. Each frequency sweep (15 logarithmically spaced points from 20 Hz to 1 MHz) was completed in ≈2 s, resulting in 600 points per sweep.

A custom-designed PCB was created with two copper contact pads for the nanomembrane surface for reliable electrical interfacing, as illustrated in Figure 1b.

All experiments were carried out under ambient laboratory conditions, without active control of temperature or humidity. This approach was justified for the nanomembranes, as previous studies [49] have shown that their electrical properties remain stable across a wide range of environmental conditions. In the case of the commercial electrode-based sensors, the decision was supported by the operational range specified in the instrumentation datasheets [66] and by the optimal working conditions reported for SPEs [67,68].

### 3.3. Preliminary Outputs

This subsection reports the initial findings of the measurement campaign.

For CV, the voltage-current curves obtained from one representative SPE per concentration level are shown in Figure 4a. The results reveal a clear correlation between analyte concentration and the amplitude of the anodic/cathodic peaks.

Similarly, the CA results in Figure 4b display time-domain current responses, plotted on a logarithmic scale for better visibility. Each curve corresponds to a specific concentration. As expected, the 0 mM condition exhibits higher noise due to the absence of reactive species, while steady-state currents strongly correlate with benzoquinone levels.

The EIS results for SPEs are shown in Figure 4c (Nyquist plots). A clear trend is observed between semicircle diameter (i.e., charge transfer resistance R_ct_) and concentration, except for the 0 mM case. This outlier highlights the limitations of classical EIS in detecting analytes at low concentrations due to minimal redox activity.

These results are consistent with expectations based on the literature [69,70], thereby confirming the anticipated behavior. It is important to emphasize once again that these techniques—albeit to varying degrees—are conventionally employed for estimating pollutant concentrations.

To qualitatively evaluate the nanomembrane response, Figure 5 shows the absolute impedance magnitude over time for four representative sensors. A sudden impedance change, with less than a few seconds of settling, occurs across all frequencies immediately following the drop at the 5-min mark, in line with the conductivity behavior described in Equation (Equation 1). It is also possible to observe a slight drift of the response over time, suggesting that it is advantageous to conduct the analyses in the first few minutes after the drop.

However, the results obtained by considering only the absolute impedance modulus of different membranes do not show a clear relationship with pollutant concentrations; this is at the expense of repeatability. By contrast, after applying the normalization strategy described in Equation (Equation 6), the results shown in Figure 6 reveal a consistent correlation between the nanomembrane’s response and different concentration levels. This normalization process increases the sensitivity of the analysis process in favor of greater repeatability of the outputs, allowing a significant relationship between the normalized impedance modulus and pollutant concentration. It has been assessed that the same behavior is observed in conventional electrode-based sensors, the results of which correlate with pollutant concentration.

Specifically, the data show a positive variation in the presence of pure water and a negative variation when pollutants are present, with the magnitude of the response increasing with concentration. Frequency analysis indicates that low frequencies yield minimal sensitivity, whereas higher frequencies produce more pronounced changes. Moreover, the separation among frequency responses following the solution drop reflects a clear dependence on pollutant quantity, justifying the introduction of the spread factor defined in Equation (Equation 7).

Overall, this normalization step not only improves reproducibility and mitigates sample variability but also highlights the diagnostic potential of nanomembranes for environmental monitoring applications.

These preliminary results confirm the feasibility of both sensor types—electrode-based sensors and nanomembranes—for pollutant concentration estimation in industrial wastewater. They lay the foundation for the subsequent development of analytical models, including both linear regression and machine-learning-based approaches, which will be discussed in the following sections.

## 4. Linear Model and Machine-Learning Approach

This section presents a detailed comparative analysis of the performance of conventional electrode-based sensors and nanomembranes. Specifically, two models will be tested: the first based on linear regression and the second employing machine-learning techniques. The models will be evaluated by comparing the error for each sensor type and assessing the accuracy of the machine-learning approach. The final part of the analysis will focus on the detection capabilities of conventional techniques and the nanomembrane-based system for identifying pollutants.

### 4.1. Linear Model Approach

This subsection aims to derive a synthetic parameter from each measurement technique from all acquisitions and establish a linear regression model between the parameter and the tested concentration.

#### 4.1.1. Model Configuration

The data collected during the monitoring campaign were used to configure a linear model, as defined in Equation (Equation 8). All these analyses were conducted in the MATLAB (R2024b version) environment.(8)$$cc=a_{0}+a_{1}\;\cdot \;P_{x}$$
where:cc is the unknown concentration;$a_{0}$ and $a_{1}$ are the linear model coefficients;$P_{x}$ is the synthetic parameter.

Data from 12 SPEs/nanostructured membranes were used, ensuring even distribution with 3 sensors per tested concentration for model configuration and 4 for model testing. A cross-validation approach was adopted to assess the model’s ability to quantify the target pollutant and to make the model more robust.

The synthetic parameters were defined as follows:For CV, the area under the current-voltage curve was used. This was calculated in MATLAB by integrating the entire curve, representing the total charge transferred during the redox reaction, which correlates directly with the analyte concentration [53].For CA, the plateau current was adopted [71,72], calculated as the average of the last 30 s of each test, since this time window, as can be observed in Figure 4b, is reasonable to be able to assert that the current has reached steady state.For EIS on electrode sensors, the chosen parameter was the difference between the mean values of the minimum peaks in the Nyquist diagrams for each concentration. This metric represents the charge transfer resistance ($R_{ct}$), which is inversely proportional to analyte concentration, as widely demonstrated in previous studies [70,73,74].For nanomembranes, EIS time-domain analysis, the response at 1 MHz was selected as the representative synthetic parameter. This value was computed as the average of the normalized signal over a 180-s window, starting 10 s after the initial impedance drop. The choice of this high-frequency component is justified by the observations discussed in Figure 6, which highlighted its high sensitivity to concentration changes. Additionally, the selected empirical times are entirely consistent with the considerations regarding signal drift and post-drop stabilization presented in Section 3.

However, initial analyses revealed that a linear model may not adequately capture the trends for some techniques—particularly EIS applied on electrode sensors and EIS applied on nanomembranes—prompting the need for more advanced modeling approaches.

Model parameters and type A uncertainties, therefore based on an a posteriori approach considering standard deviations, evaluated through Matlab functions “polyfit” and “polyval”, are reported in Table 3. These preliminary results highlight that EIS on electrode-based sensors and EIS on nanomembranes at 1 MHz are not well-suited to linear regression, suggesting the need for alternative strategies to capture the relationship between sensor output and analyte concentration.

#### 4.1.2. Model Test

The performance of a linear regression model, expressed as the error between estimated and actual concentrations, is reported in Table 4, averaged across four cross-validation folds. The results indicate that CV and CA provide relatively consistent estimates, whereas electrode-based EIS and nanomembrane EIS exhibit large errors, in line with the uncertainties reported in Table 3. Moreover, the variability in the nanomembrane response across concentrations, as shown in Figure 7, reveals that the nanomembrane outputs are overlapped and therefore indistinguishable within a regression framework to determine the pollutant concentration. These findings clearly demonstrate that a conventional linear regression approach is inadequate for nanomembranes due to their non-linear response and their poor reproducibility.

Given these limitations, a different analytical paradigm was required. Rather than enforcing a regression model, the problem was reformulated as a classification task using machine-learning methods. This transition from regression to classification enables the use of models inherently better suited to handling the variability and nonlinearity of nanomembrane responses. As will be shown in the following section, this approach effectively mitigates the shortcomings of the linear model, allowing nanomembrane sensors to achieve reliable discrimination of pollutant concentrations when coupled with ML-based classifiers.

### 4.2. Machine-Learning Approach

Based on the experimental campaign, the acquired dataset for each technique was organized for training and testing. The training set included data from 12 SPEs/nanostructured membranes (3 per concentration), as reported in Table 5. A 5% subset of this training data was set aside for validation using a 5-fold cross-validation scheme. The test set comprised measurements from the 4 remaining sensors, each corresponding to a different concentration level, as shown in Table 6. Each concentration was treated as a distinct class, resulting in a total of four classes.

The data points considered and the final dataset dimensions for each technique are, respectively, reported in Table 7 and Table 8. Feature extraction was tailored to the nature of each technique:CV features consisted of both voltage and current at each excitation level;CA included the full current–time profile;EIS from electrode-based sensors comprised magnitude, phase, and the real and imaginary impedance components across all 35 measured frequencies, given that real and imaginary parts are not directly correlated to magnitude and phase in the equivalent circuit model [70,75];EIS for nanomembranes in the time domain included 15 normalized frequency responses and a spread factor (see Equations (Equation 6) and (Equation 7)), extracted from the first 180 s following pollutant addition (excluding the initial 10 s to ensure a minimum settling time). The baseline was defined as the average of the first 5 min to guarantee sensor stability. The choice of the 3-min window reflects the fact that the most informative signal dynamics occur immediately after pollutant addition. At the same time, longer acquisition periods may introduce drift caused by external disturbances (e.g., evaporation or environmental contaminants) not directly related to the interaction with the target molecules. If one had trained the model with all available data, including the drift, it would have been trained on potentially erroneous data, so 3 min is an optimal balance between volume and stability of data.

At this stage, no feature selection was applied.

Machine-learning classification was conducted using the MATLAB *Classification Learner* toolbox [76], which provides a standardized and reproducible environment for model training, automatic hyperparameter optimization, and cross-validation. The study considered four supervised algorithms, selected to represent different families of classifiers after preliminary evaluations available in the toolbox:
Cubic SVM. A support vector machine with a third-degree polynomial kernel. The normalization of the data was enabled, and the kernel scaling together with the box constraint from which the support vectors derive were automatically optimized by the toolbox (the software uses a heuristic procedure to select the parameters).Subspace k-NN. An ensemble of k-nearest neighbors learners trained on selected feature subspaces. The number of learners and the subspace dimensions were determined by the toolbox optimization procedure, while Euclidean distance was used as the similarity metric.Efficient Logistic Regression. The regularization strength was adjusted automatically, ensuring an appropriate trade-off between bias and variance. The Auto setting sets lambda equal to 1/*n*, where *n* is the number of observations in the training sample. The solver used was SGD for efficient logistic regression. While the regularization strength (lambda) (or the number of in-fold observations, if using cross-validation).Kernel Naive Bayes. A probabilistic classifier using Gaussian kernel density estimation. Bandwidth parameters for the kernel were optimized through the toolbox’s built-in procedure.

Each method is briefly summarized below:SVM is a supervised max-margin model with associated learning algorithms that analyze data by finding the optimal hyperplane that maximizes the margin between classes;KNN assigns classes based on the closest training samples, using different metrics;Logistic Regression models class probabilities using a sigmoid function; ELR is a computationally efficient variant implemented in MATLAB;KNB combines kernel density estimation with Bayes’ theorem, offering robustness in small datasets and good classification accuracy.

Classification performance was evaluated using accuracy, precision, recall, F1-score, and ROC-AUC, as reported by the toolbox [76].

To avoid potential confusion with terminology used in traditional chemical techniques, it is important to clarify that within the context of machine learning, classification of concentration is equivalent to determination of concentration.

The results were evaluated using confusion matrices whose accuracies were tabulated in Figure 8 for greater readability of the results.

The findings reveal that CA and EIS on nanomembranes outperform other techniques. In particular:CV does not lend itself well to classification, as all tested algorithms yielded poor performance. Only the k-Nearest Bayesian (KNB) classifier achieved a moderate average accuracy level; however, even in this case, the classification accuracy for the 0.1 mM class remained below 50%.CA clearly emerges as the most robust technique, consistently delivering outstanding results across nearly all algorithms. With the sole exception of the Support Vector Machine (SVM), all models achieved an average accuracy exceeding 99%, with class-wise accuracies surpassing 98% across all valid algorithms.EIS on electrode sensors proves to be the most challenging technique, as anticipated from the preliminary analysis in Section 3. None of the tested algorithms demonstrated satisfactory performance in this context.EIS on nanomembranes, by contrast, exhibited good average classification performance—above 60% accuracy—with all algorithms except SVM. Nonetheless, certain classes failed to exceed 50% accuracy when classified using KNN and ELR. However, KNB yielded exceptional results: each class was accurately identified, achieving 100% accuracy in all but the 1 mM class, which still reached over 75% accuracy. The overall average KNB accuracy exceeded 90%.

These results highlight several critical insights.

CA emerges as an exceptionally robust electrochemical technique, consistently yielding superior classification performance across both linear and machine-learning-based models. Its reliability and effectiveness are further reinforced by its long-standing status as the most established and widely adopted method for quantifying chemical species.

In addition, the integration of machine-learning models reveals that even nanomembrane-based systems—limited by the previously tested linear approaches—can achieve high levels of accuracy and reliability in pollutant detection. This demonstrates that the proposed nanosensors represent a viable and promising alternative to conventional techniques and commercial electrode-based sensors, with the potential to deliver comparable, and in some cases superior (considering CV and EIS on electrode sensors), analytical performance.

### 4.3. Sensitivity Performance Evaluation over Selectivity

The previous analysis reveals two key insights:
ML significantly improves the performance of nanomembrane-based EIS over linear modeling;Only the KNB algorithm provides acceptable classification accuracy for nanomembranes.

Notably, few false positives were observed in nanomembranes in the 0 mM class, suggesting that while concentration estimation is difficult, pollutant detection is more feasible.

Based on this, the classification task has been restructured to focus on detection (pollutant sensitivity) rather than quantification (pollutant selectivity). The classes were reduced to clean water (0 mM) and polluted water (0.1, 1, and 10 mM). Since the newly defined polluted water class (aggregating all concentrations) was oversized compared to the pure water class, an equal number of points per concentration was randomly selected to match the pure water class. In this way, the dataset was organized balanced across all classes, replicating the distribution and structure of the original 4-class dataset. The considered data points are reported in Table 9 and the total size of the dataset used in this study is reported in Table 10.

Confusion matrices for the 2-class task were analyzed, and results were summarized in the same way as the previous section in Figure 9. This binary classification demonstrated excellent results for nanomembranes, achieving 100% accuracy across all algorithms (excluding KNB). CA also maintained high performance, while classical EIS remained the least effective. CV, with ML, showed acceptable results ( 70% accuracy) only when using KNB.

## 5. Extensive Nanomembranes Performance Analysis with ML Approach

Building upon the results discussed before, this section delves deeper into the innovation and potential of the nanosensors. In particular, performance is evaluated as two operational parameters change: test duration and noise robustness, to assess nanomembranes’ performance under conditions closely resembling real-time field scenarios.

### 5.1. Test Duration: Dataset Size

Datasets of varying sizes were considered to replicate conditions as closely as possible to real-time water monitoring and analysis. Starting from the initial full dataset, both the training and test sets were subdivided and used to evaluate classification accuracy for both 4-class and 2-class configurations. This approach reflects the relationship between dataset size and the observation window, directly influencing analysis time.

The previously examined 180-s interval, beginning 10 s after the solution was dropped, was divided into equal subintervals to generate different observation windows. The time windows analyzed included: 180, 150, 120, 90, 45, 30, and 15 s. The resulting datasets, corresponding to both 4-class and 2-class configurations, are reported in Table 11 and Table 12, respectively. Starting from the 180-s condition, each subsequent interval represents a reduction in observation time and, consequently, in dataset size. In this stage of work, the baseline was defined as the average of the first 30 s of the test to decrease the time considered.

Training and test sets were organized following the previously established protocol, with 5% of the training data withheld for cross-validation. The analysis employed the most effective classifiers identified in the prior section: KNN for the 2-class configuration, and KNB for the 4-class classification task.

Selecting different dataset sizes allows for identifying both the optimal training duration and the minimum required testing window for accurate classification. From an operational point of view, classifier training could typically be performed offline, where timing constraints are less stringent. Therefore, the optimal training time is defined as the duration required to achieve maximum classification accuracy. Conversely, the testing observation window should be minimized to allow rapid detection in the field. The first phase of the analysis assessed classifier accuracy as a function of the observation window. The results for the 4-classes configuration, shown in Figure 10a, indicate that the highest performance is achieved when the model is trained on longer time windows and tested on shorter ones. Specifically, the best configuration for the 4-class quantifier task involves training on 90 s and testing on 15 s. In contrast, for the 2-class pollutant detector, maximum performance is consistently achieved across all time windows, as observed in Figure 10b, including the shortest (15 s for both training and testing).

These results highlight that pollutant detection does not require extended observation time—the system responds almost instantaneously. For quantification, however, a clear trend is observed in the Figure 10a: the longer the training window, the better the system’s ability to discriminate among concentration levels.

This confirms the potential for a dual-use approach:
A rapid, real-time response for pollutant detection, feasible within a few seconds;A more detailed, offline quantification analysis, with cloud-based training performed within minutes after detection.

### 5.2. Noise Analysis

With the optimal training duration (15 s for detection and 90 s for quantification) and testing window (15 s) based on the criteria chosen earlier, the robustness of the system to noise was assessed. To this end, Gaussian additive noise with zero mean was progressively introduced into the test set, while the training set remained unaltered. Unlike laboratory environments, this simulates field conditions, where environmental noise is inherently present and cannot be entirely eliminated. Noise levels were set at 5, 10, 25, 35, and 50 times the standard deviation of the baseline condition, calculated for each frequency and membrane. For reference, the worst-case standard deviation under baseline conditions was 0.52 Ω. Figure 11 shows the confusion matrices for both classifiers (KNN for 2 classes, KNB for 4 classes) across increasing noise levels. The noise level, according to the above, is defined as [5σ, 10σ, 25σ, 35σ, 50σ].

The results demonstrate strong noise resilience for both classifiers, with the 2-class system exhibiting superior robustness. Up to 5σ of noise, classification accuracy remains unchanged for both models. At 10σ and 25σ, minor degradation appears in the 4-class classifier, particularly affecting the 0 mM and 10 mM classes. Beyond 35σ, degradation becomes evident across all classes for the 4-class model. Conversely, the 2-class classifier exhibits only negligible performance loss even at the maximum noise level (50σ), confirming exceptional robustness. A summary of average classification accuracies at each noise level is provided in Table 13.

## 6. Conclusions

This study presents an innovative methodology for the detection and quantification of water pollutants in industrial wastewater using nanomembrane-based sensors in conjunction with electrochemical impedance spectroscopy. The proposed approach was validated through a comprehensive performance evaluation compared to conventional electrochemical techniques—cyclic voltammetry, chronoamperometry, and electrochemical impedance spectroscopy—using commercial electrode-based sensors.

The validation experiments were conducted using benzoquinone, a representative organic industrial pollutant, at four different concentration levels tested: 0 mM, 0.1 mM, 1 mM, and 10 mM. The study explored the most suitable modeling approaches for interpreting the nanomembrane sensor responses in pollutant quantification tasks. Initial analyses using linear models revealed superior performance of conventional techniques, particularly CV and CA. However, when applying machine-learning approaches, the nanomembrane-based sensors demonstrated significantly improved reliability, achieving performance levels comparable to, and in some cases exceeding, those of traditional methods.

In quantification tasks, nanomembranes achieved classification accuracies of up to 94%, approaching the 99% accuracy achieved by CA with commercial sensors. Nanomembranes performed even more impressively in detection (binary classification) tasks, reaching up to 100% accuracy, confirming their high potential as selective detectors.

A key objective of the study was also to assess the real-time and in situ applicability of the proposed system. This was achieved by examining different dataset sizes and introducing various noise levels, respectively. The nanomembrane-based system maintained robust performance: in 4-class classification tasks, accuracy remained above 85% even with 15σ added noise, while in the 2-class detection scenario, 100% accuracy was preserved up to 25σ noise (standard deviation), and 96% accuracy was retained even under the highest noise conditions tested.

From a system design perspective, critical insights emerged regarding time constraints. For quantification, longer training time windows (optimal at 90 s) increased accuracy while maintaining a fixed test duration of 15 s. For detection tasks, only 15 s of signal acquisition were required for training, enabling real-time alerting and rapid response capabilities. This dual-mode functionality supports a scalable deployment strategy: fast on-site detection, followed by more detailed quantification either in the cloud or offline.

The study also highlighted a key trade-off: detection tasks are inherently more robust, while quantification—particularly under high noise or limited acquisition time—is more sensitive. Nonetheless, the overall performance remains satisfactory and aligned with the requirements of early-stage field deployment.

For future work, several promising directions will be pursued to develop further and consolidate the proposed methodology:Validation on real-world water samples, including complex industrial and environmental matrices;Chemical functionalization of nanomembranes to enhance selectivity toward specific analytes;Extension of the study to additional pollutants and concentration ranges, with a more refined investigation between 0.1–1 mM and 1–10 mM;Adoption of ML-based regression algorithms, rather than classifiers, to evaluate intermediate values within the newly tested concentration ranges;Cost estimation and a large-scale integration plan.

Finally, future work will particularly focus on the design of a low-cost and portable hardware platform for wastewater pollutant monitoring, integrating the investigated nanomembranes. Currently, the technology remains at a prototype level, tested only under simplified laboratory conditions. The planned hardware development aims to transform this proof-of-concept into a robust, field-deployable system capable of operating within complex industrial wastewater matrices, thereby enabling reliable real-time in situ monitoring. In conclusion, this study establishes a solid first step toward nanomembrane-based pollutant sensing. At this prototype stage, the approach demonstrates promising performance when coupled with ML, achieving accuracy levels comparable to those of traditional electrochemical methods. These findings represent a significant advancement toward real-time, innovative wastewater quality monitoring systems with strong potential for scalable industrial deployment.

## Figures and Tables

**Figure 1 sensors-25-05390-f001:**
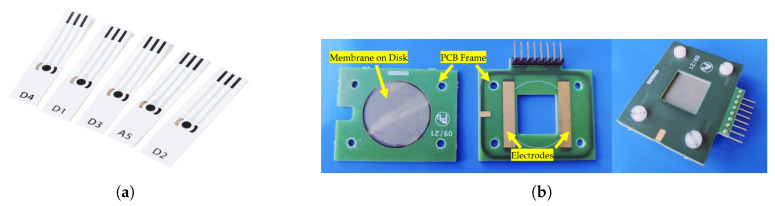
Sensors used in this study: (**a**) commercial SPEs (image sourced from the ItalSens website [52]); and (**b**) nanomembranes and custom PCB interface circuit for signal acquisition.

**Figure 2 sensors-25-05390-f002:**
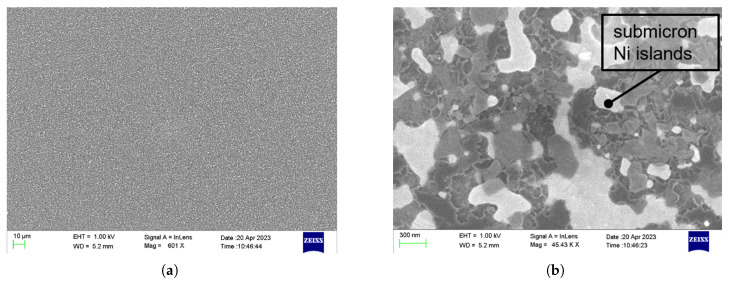
SEM images of the fabricated films: (**a**) with the magnification of 601; and (**b**) magnification 45,000.

**Figure 3 sensors-25-05390-f003:**
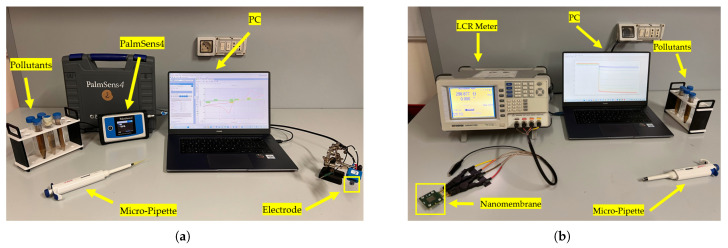
Experimental setup: (**a**) for CV, CA, and EIS on commercial electrode-based sensors; (**b**) for time-domain EIS on nanomembranes.

**Figure 4 sensors-25-05390-f004:**
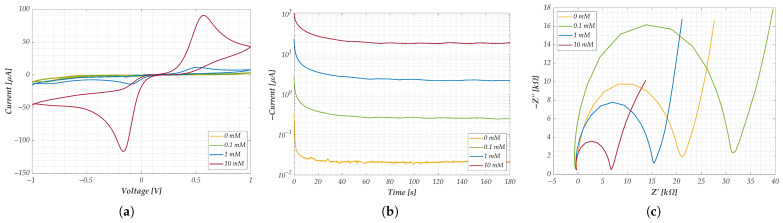
Preliminary results obtained for each technique at different concentrations: (**a**) CV; (**b**) CA; and (**c**) EIS. For CA, results are reported on a logarithmic scale for greater readability. The different colors used for the curves indicate the concentrations tested.

**Figure 5 sensors-25-05390-f005:**
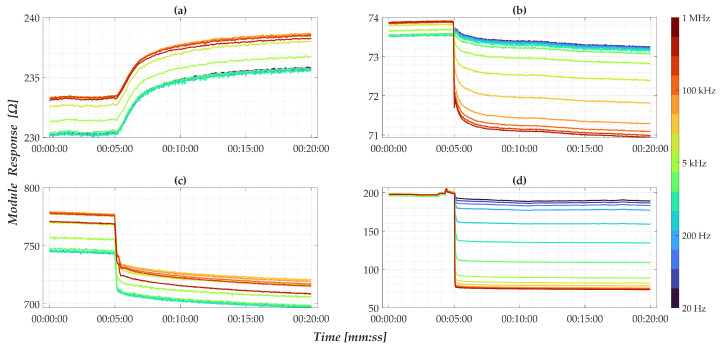
Absolute time-domain impedance response of nanomembranes at (**a**) 0 mM; (**b**) 0.1 mM; (**c**) 1 mM; and (**d**) 10 mM. The color scale indicated refers to the different analysis frequencies tested.

**Figure 6 sensors-25-05390-f006:**
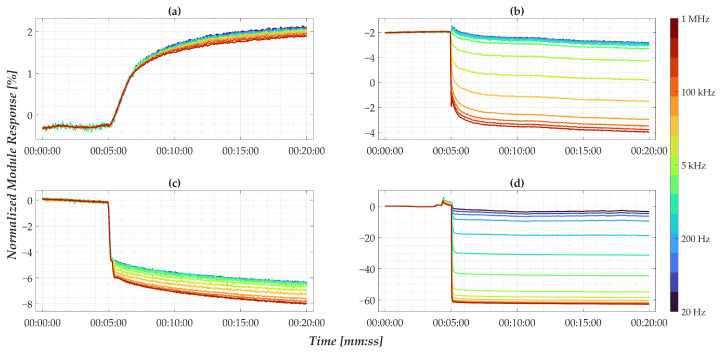
Normalized time-domain impedance response of nanomembranes at (**a**) 0 mM, (**b**) 0.1 mM, (**c**) 1 mM, and (**d**) 10 mM. The color scale indicated refers to the different analysis frequencies tested.

**Figure 7 sensors-25-05390-f007:**
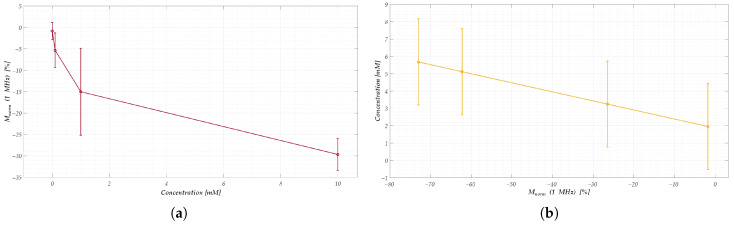
Reproducibility and limitations of the linear regression model for nanomembranes: (**a**) average normalized response $M_{norm}\;\left(1\;\mathrm{MHz}\right)$ at different concentrations with relative standard deviation with a confidence level of 99.7 %, used for model construction; (**b**) validation of the regression model with the associated uncertainty. The results highlight the strong overlap among concentration responses and the incompatibility of a linear regression framework for pollutant quantification.

**Figure 8 sensors-25-05390-f008:**
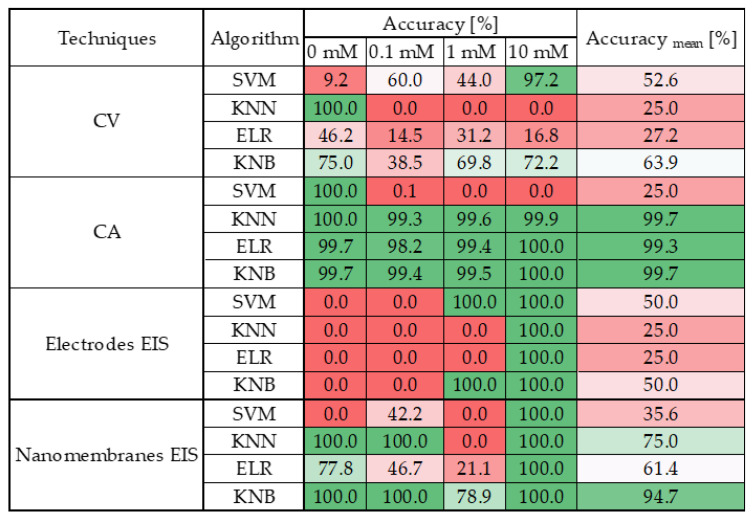
Accuracy of the classification obtained from the conducted analyses. In particular, it reports, for each measurement technique and for each algorithm tested, the percentage of accuracy for each concentration tested and the total average accuracy. A color map has been added to make the matrix easier to understand, with green representing maximum accuracy and red representing minimum accuracy.

**Figure 9 sensors-25-05390-f009:**
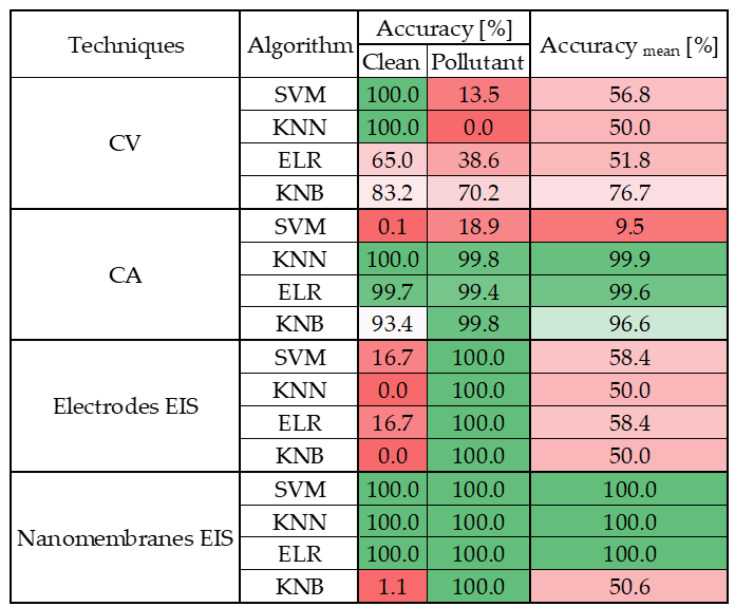
Classification accuracy using a 2-class model (detection-only). In particular, it reports, for each measurement technique and algorithm tested, the accuracy percentage for each class tested and the total average accuracy. A color map has been added to make the matrix easier to understand, with green representing maximum accuracy and red representing minimum accuracy.

**Figure 10 sensors-25-05390-f010:**
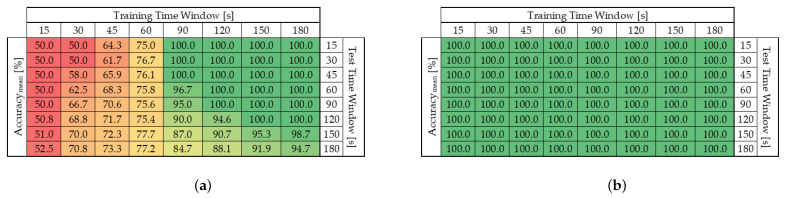
Classification accuracy across different observation windows: (**a**) 4-class quantification, and (**b**) 2-class detection. In particular, the average accuracy percentage obtained for each training and test time is reported. A color map has been added to make the matrix easier to understand, with green representing maximum accuracy and red representing minimum accuracy.

**Figure 11 sensors-25-05390-f011:**
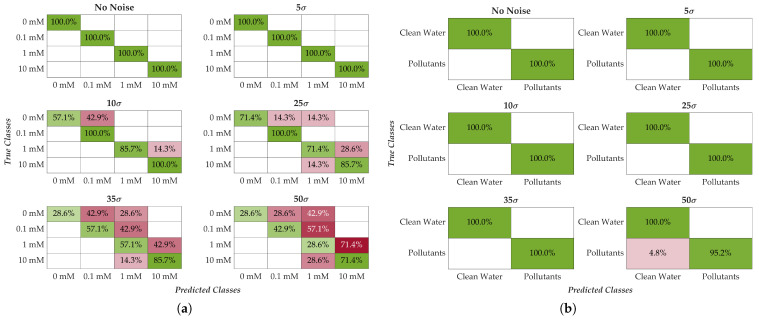
Accuracy of classification with respect to different noise levels considered. Shows the confusion matrices as the noise levels increase: (**a**) 4-class quantification and (**b**) 2-class detection. Noise levels considered are [0, 5, 10, 25, 35, 50] times the standard deviation (σ). A color map has been added to make the matrix easier to understand, with green representing maximum accuracy and red representing minimum accuracy.

**Table 1 sensors-25-05390-t001:** Comparison of the analyzed sensing technologies [16].

Technology	Real-Time Capability	Portability	Limitations
Colorimetric [17,18,19,20,21,22,23,24,25,26,27,28,29,30]	Limited	Very high	Color, turbidity and lighting interferences; long-term stability.
Lab-on-a-Chip [31,32,33,34,35,36,37,38,39]	High	High	Fabrication process complexity, costs
Raman/SERS [40,41,42]	Moderate	Moderate	Fluorescence interference; costly instrumentation.

**Table 2 sensors-25-05390-t002:** Comparison of sensing technologies from selected studies and reviews.

Technology	Real-Time Capability	Portability	Limitations
Electrochemical ISM [43]	High	High	pH/O_2_ interferences, fouling
Thin-film ISM [44]	High	High	T/pH corrections, limited lifetime
Tryptophan-like fluorescence probe [45]	High	High	Calibration, particles interferences, cost
Optical fluorescence/UV sensors [46]	Moderate	Moderate	Turbidity, fouling
Electrochemical sensors [46]	High	High	Fouling
Spectroscopy & chromatography [46]	Low	Low	Cost, slow response
IoT wireless multiparameter systems [47]	High	High	Energy consumption, reliability
Enzymatic biosensors [47]	High	High	Biological stability, regeneration
Integrated multisensor with LoRa transmission [48]	High	High	Calibration, network dependency

**Table 3 sensors-25-05390-t003:** Linear model parameters and associated model uncertainty for each tested technique.

Technique	$a_{0}$ [mM]	$a_{1}$	$u_{\mathrm{mod}}$ [mM]
CV	2.77	4.36 [mM/C]	0.14
CA	2.77	−4.15 [mM/μA]	0.30
Classical EIS	3.70	−1.87 [mM/Ω]	3.16
$M_{norm}\;\left(1\;\mathrm{MHz}\right)$	2.75	−2.09 [mM/%]	2.48

**Table 4 sensors-25-05390-t004:** Estimation error for each tested technique across all concentrations.

Concentration [mM]	CV Error [mM]	CA Error [mM]	EIS Error [mM]	$M_{\mathrm{norm}}$ (1 MHz) Error [mM]
0	−0.09	−0.08	4.62	0.83
0.1	0.01	−0.03	3.13	0.76
1	0.11	0.26	4.14	−0.34
10	−0.02	−0.07	−2.16	−8.01

**Table 5 sensors-25-05390-t005:** Distribution of training data of SPEs and nanomembranes per class (each sensor is represented by index).

SPE Index	Nanomembrane Index	Class
1, 2, 3	1, 2, 3	0 mM
4, 5, 6	4, 5, 6	0.1 mM
7, 8, 9	7, 8, 9	1 mM
10, 11, 12	10, 11, 12	10 mM

**Table 6 sensors-25-05390-t006:** Distribution of test data for SPEs and nanomembranes per class (each sensor is represented by index).

SPE Index	Nanomembrane Index	Class
13	13	0 mM
14	14	0.1 mM
15	15	1 mM
16	16	10 mM

**Table 7 sensors-25-05390-t007:** Distribution of data points per SPE and per nanomembrane to build the 4 classes od dataset.

Technique	Data Points
CV	400 × 2/SPE
CA	1800 × 1/SPE
Electrode Sensors EIS	20 × 140/SPE
Nanomembranes EIS	90 × 16/Nanomembrane

**Table 8 sensors-25-05390-t008:** Dataset size (rows × features) for training and test sets considering 4 classes.

Dataset	CV	CA	Electrode Sensors EIS	Nanomembranes EIS
Training Data	4800 × 2 1200 × 2/class	21600 × 1 5400 × 1/class	240 × 140 60 × 140/class	1080 × 16 270 × 16/class
Test Data	1600 × 2 400 × 2/class	7200 × 1 1800 × 1/class	80 × 140 20 × 140/class	360 × 16 90 × 16/class

**Table 9 sensors-25-05390-t009:** Distribution of data points per SPE and per nanomembrane considering 2 classes.

Technique	Data Points
CV	400 × 2/SPE for clean water class 133 × 2/SPE for polluted water class
CA	1800 × 1/SPE for clean water class 600 × 1/SPE for polluted water class
Electrode Sensors EIS	20 × 40/SPE for clean water class 7 × 40/SPE for polluted water class
Nanomembranes EIS	90 × 16/SPE for clean water class 30 × 16/SPE for polluted water class

**Table 10 sensors-25-05390-t010:** Dataset size (rows × features) for training and test sets considering 2 classes.

Dataset	CV	CA	Electrode Sensors EIS	Nanomembranes EIS
Training Data	2400 × 2 1200 × 2/class	10800 × 1 5400 × 1/class	120 × 140 60 × 140/class	540 × 16 270 × 16/class
Test Data	800 × 2 400 × 2/class	3600 × 1 1800 × 1/class	40 × 140 20 × 140/class	180 × 16 90 × 16/class

**Table 11 sensors-25-05390-t011:** Dataset dimensions for each observation window considering 4 classes.

Dataset	180 s	150 s	120 s	90 s	60 s	45 s	30 s	15 s
Training	1080 × 16	900 × 16	720 × 16	540 × 16	360 × 16	270 × 16	180 × 16	90 × 16
Test	360 × 16	300 × 16	240 × 16	180 × 16	120 × 16	90 × 16	60 × 16	30 × 16

**Table 12 sensors-25-05390-t012:** Dataset dimensions for each observation window considering 2 classes.

Dataset	180 s	150 s	120 s	90 s	60 s	45 s	30 s	15 s
Training	540 × 16	450 × 16	360 × 16	270 × 16	180 × 16	135 × 16	90 × 16	45 × 16
Test	180 × 16	150 × 16	120 × 16	90 × 16	60 × 16	45 × 16	30 × 16	15 × 16

**Table 13 sensors-25-05390-t013:** Classification accuracy across different noise levels.

Noise Level	Accuracy (4 Classes) [%]	Accuracy (2 Classes) [%]
0	100	100
5σ	100	100
10σ	85	100
25σ	82	100
35σ	57	100
50σ	42	96

## Data Availability

The raw data supporting the conclusions of this article will be made available by the authors on request.

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
