# Peer review of "A Novel ML-Powered Nanomembrane Sensor for Smart Monitoring of Pollutants in Industrial Wastewater"

_sensors, 2025, doi:10.3390/s25175390_

Round 1
Reviewer 1 Report
Comments and Suggestions for Authors
Recommendation: Major Revision
- The article claims novelty (nanomembrane sensor + AI) but does not quantitatively compare its performance with existing sensors in the literature. The contribution's positioning relative to the state of the art in industrial pollutant detection remains superficial.
- Add a detailed comparison table (performance, limitations, costs) with existing approaches.
- Clarify the original contribution (nanostructure, ML algorithm, IoT integration) and justify why it outperforms current methods.
- The physical and chemical properties of the nanomembrane (composition, thickness, pore size, chemical stability) are poorly detailed. There is no clear indication of manufacturing reproducibility or production tolerances.
- Provide complete structural parameters, accompanied by SEM images.
- Discuss reproducibility and variability between production batches.
- The description of the ML algorithm is vague (model type, hyperparameters, stopping criteria, optimization). The size, diversity, and origin of the dataset are not specified; there is a risk of overfitting if the dataset is limited.
- Precisely describe the ML architecture, hyperparameters, cross-validation method, and metrics used.
- Specify the size and representativeness of the training and test data, and justify their relevance for real-world industrial applications.
- The tests appear to be conducted under ideal laboratory conditions, without demonstration in a real industrial environment. Potential interferences (pH, temperature, pollutant mixture) are not tested.
- Conduct tests on real industrial effluents, with varying conditions and the presence of multiple contaminants.
- Include robustness and sensor lifespan tests.
- The results (accuracy, sensitivity) are provided without confidence intervals or statistical analysis. There is no comparison of performance with and without ML integration to demonstrate the real impact of AI.
- Provide confidence intervals or statistical tests to validate the significance of performance.
- Compare the raw sensor performance with that optimized by ML to quantify the real contribution of AI.
- No discussion of manufacturing, maintenance, or plant integration costs. No consideration of compliance with environmental regulations.
- Add a cost estimate and a large-scale integration plan.
- Discuss compliance with industrial wastewater monitoring standards.
Here are some articles you should use and cite (all) to improve your article (note that these references are not part of my articles):
- Gu, K., Liu, Y., Liu, H., Liu, B., Qiao, J., Lin, W.,... Zhang, W. (2025). Air Pollution Monitoring by Integrating Local and Global Information in Self-Adaptive Multiscale Transform Domain. IEEE Transactions on Multimedia, 27, 3716-3728. doi: 10.1109/TMM.2025.3535351
- Zhang, L., Liu, Y., Zeng, G., Yang, Z., Lin, Q., Wang, Y.,... Pu, S. (2023). Two-dimensional Na-Bentonite@MXene composite membrane with switchable wettability for selective oil/water separation. Separation and Purification Technology, 306, 122677. doi: https://doi.org/10.1016/j.seppur.2022.122677
- Pan, Y., Liu, H., Huang, Z., Zhang, W., Gao, H., Liang, L.,... Meng, H. (2024). Membranes based on Covalent Organic Frameworks through Green and Scalable Interfacial Polymerization using Ionic Liquids for Antibiotic Desalination. Angewandte Chemie International Edition, 63(4), e202316315. doi: https://doi.org/10.1002/anie.202316315
- Qiao, Y., Lü, J., Wang, T., Liu, K., Zhang, B.,... Snoussi, H. (2024). A Multihead Attention Self-Supervised Representation Model for Industrial Sensors Anomaly Detection. IEEE Transactions on Industrial Informatics, 20(2), 2190-2199. doi: 10.1109/TII.2023.3280337
- Hu, C., Zhao, C., Shao, H., Deng, J., & Wang, Y. (2024). TMFF: Trustworthy Multi-Focus Fusion Framework for Multi-Label Sewer Defect Classification in Sewer Inspection Videos. IEEE Transactions on Circuits and Systems for Video Technology, 34(12), 12274-12287. doi: 10.1109/TCSVT.2024.3433415
- Fan, W., Xin, Q., Dai, Y., Chen, Y., Liu, S., Zhang, X.,... Gao, X. (2025). Competitive transport and adsorption of CO2/H2O in the graphene nano-slit pore: A molecular dynamics simulation study. Separation and Purification Technology, 353, 128394. doi: https://doi.org/10.1016/j.seppur.2024.128394
Reviewer 2 Report
Comments and Suggestions for Authors
This study proposes a graphitic nanomembrane sensor with time-domain EIS and machine learning for industrial pollutant detection. Experimental validation using benzoquinone demonstrates that ML algorithms (especially KNB) significantly enhance nanomembrane performance, achieving >94% classification accuracy and 100% detection robustness under noise. I think this work is interesting, but before it is published, the following issues need to be addressed:
1) It is suggested that the authors supplement important quantitative conclusions in the introduction section. For instance, the authors mentioned that the combination of nanosensors and ML algorithms brought about a "significant performance improvement". Among them, "significant" needs to be quantified to highlight the advantages of the research results.
2) In the introduction, references [9-11] mention "multiple efficient water monitoring systems", but do not specify their key indicators, such as detection limits, response times, and accuracy rates. Has the "efficient system" in reference [9] achieved an accuracy rate of 99% under similar pollutants (benzoquinone)? What is the actual detection limit of commercial sensors (such as ItalSens SPE) in the original reference [17]? If no quantitative benchmark is established, it is unconvincing to claim that the nanomembrane has "significantly improved performance" (abstract) or "overcome current technical limitations" (introduction). In addition, suggest the authors refer to literature of microfluidic technology 10.1016/j.snb.2023.135210, to enrich the research background.
3) In the theoretical model section, the authors proposed a transport equation based on the generalized Ohm's law. The physical meanings of formulas (1)-(3) seem to contradict the abrupt response shown in Figure 4: the theoretical model describes steady-state behavior, while the experimental observation is a transient response. In this regard, authors need to clarify the correlation between transient responses and the theoretical framework.
4) In the actual detection, the authors only used the benzoquinone/buffer system. However, actual industrial wastewater contains complex matrices, multiple ions and other interfering substances. Will this approach undermine the reliability of the claim of "real-time in-situ monitoring"?
5)Conventional EIS shows 4.62mM error (Table 2), but root cause. No optimization attempts (e.g., feature engineering) for commercial sensors weaken nanomembrane superiority argument.
6) The highest concentration of benzoquinone tested in this article is 10mM. Can this be regarded as the concentration of benzoquinone in typical industrial wastewater? If the concentration of benzoquinone in typical industrial wastewater is relatively high, is it necessary to add signal saturation or nonlinear response at high concentrations in this paper?
Round 2
Reviewer 1 Report
Comments and Suggestions for Authors
The authors have adequately addressed my recommendations, and the manuscript has been improved. However, before acceptance, I would like to request a minor revision. Specifically, references [9–14] should be properly integrated into the corrected version and not grouped together within a single sentence.
Once this adjustment is made, the manuscript will be suitable for acceptance.
Author Response
Comments 1: The authors have adequately addressed my recommendations, and the manuscript has been improved. However, before acceptance, I would like to request a minor revision. Specifically, references [9–14] should be properly integrated into the corrected version and not grouped together within a single sentence.
Once this adjustment is made, the manuscript will be suitable for acceptance.
Response 1: We agree with the reviewer and thank him for the comment. As suggested, references [9-14] have been incorporated into the text by expanding the discussion and adding more details.
Reviewer 2 Report
Comments and Suggestions for Authors
The authors have solved my problem. This article can be published!
Author Response
We thank the reviewer for the support.